# DESIRE: Dynamic Knowledge Consolidation for Rehearsal-Free Continual Learning

## Abstract

Continual learning aims to equip models with the ability to retain previously learned knowledge like a human. Recent work incorporating Parameter-Efficient Fine-Tuning has revitalized the field by introducing lightweight extension modules. However, existing methods usually overlook the issue of information leakage caused by the fact that the experiment data have been used in pre-trained models. Once these duplicate data are removed in the pre-training phase, their performance can be severely affected. In this paper, we propose a new LoRA-based rehearsal-free method named **DESIRE**. Our method avoids imposing additional constraints during training to mitigate catastrophic forgetting, thereby maximizing the learning of new classes. To integrate knowledge from old and new tasks, we propose two efficient post-processing modules. On the one hand, we retain only two sets of LoRA parameters for merging and propose dynamic representation consolidation to calibrate the merged feature representation. On the other hand, we propose decision boundary refinement to address classifier bias when training solely on new class data. Extensive experiments demonstrate that our method achieves state-of-the-art performance on multiple datasets and strikes an effective balance between stability and plasticity. Our code will be publicly available.

## 1 Introduction

Despite the remarkable achievements, AI is still far from becoming truly human-like in intelligence. Continual learning (CL) aims to address *catastrophic forgetting* (Kirkpatrick et al., 2017; Li & Hoiem, 2017; Verwimp et al., 2023), where AI systems tend to forget previously learned tasks as they learn new ones. CL encompasses both task-incremental learning (TIL) and class-incremental learning (CIL) scenarios (Wang et al., 2024), where the former allows the model to identify which task the test samples belong to during inference, while the latter requires the model to recognize all seen classes without knowing the task identity. This paper focuses on the more challenging setting of rehearsal-free CIL (Zhu et al., 2021; Liang & Li, 2024), where the model can only access the training data of the current task at each stage.

Recently, with the widespread use of pre-trained models and various parameter-efficient fine-tuning (PEFT) (Houlsby et al., 2019; Hu et al., 2021; Jia et al., 2022) methods, the field of CIL has seen rapid advancements. On the one hand, pre-trained models provide good generalization capabilities, making them naturally better than training from scratch in terms of performance. On the other hand, PEFT methods such as LoRA (Hu et al., 2021) and Prompt (Jia et al., 2022) achieve results comparable to the full fine-tuning with a significantly small number of parameters, making the application of CIL methods possible. For example, L2P (Wang et al., 2022c) and DualPrompt (Wang et al., 2022b) combine prompt tuning with pre-trained models and achieve remarkable performance. O-LoRA (Wang et al., 2023) and InfLoRA (Liang & Li, 2024) demonstrate the superiority of LoRA on CIL tasks. In addition, LAE (Gao et al., 2023) integrates Adapter, Prompt, and LoRA for CIL. All of these methods highlight the potential for applying PEFT to CIL tasks.

Nevertheless, we observe that despite such high performance achieved by these methods, two crucial issues remain: (1) *The performance is highly dependent on information leakage.* Several studies (Kim et al., 2022; Liu et al., 2023; Lin et al., 2024) have pointed out that in the absence of strong supervised pre-trained weights (e.g., ImageNet-21k), the performance of all the existing methods suffers from varying degrees of degradation, and even lower than methods not designed for PEFT.

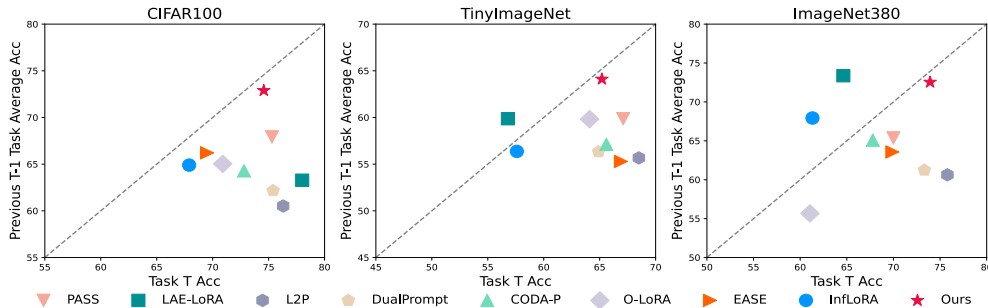

Figure 1: Stability and plasticity analysis. We visualize the accuracy of the final task ($Acc_T$) and the average accuracy of the previous $T-1$ tasks ($\frac{1}{T-1}\sum_{t=1}^{T-1} Acc_t$) at the last stage ($T = 10$) for different methods under three datasets. Methods that are closer to the diagonal and nearer to the upper-right corner of the graph are superior. More detailed results can be seen in Sec 4.4.

This phenomenon can be attributed to information leakage due to classes overlap between the experiment data in CIL and the data used in pre-trained model (Kim et al., 2023), which indirectly boosts the model's performance. (2) *Lack of balance between stability and plasticity.* Intuitively, a robust CIL system should neither sacrifice performance on new tasks to retain knowledge of old ones, nor forget old tasks excessively in order to learn new ones. However, as shown in Fig. 1, we revisit the stability and plasticity and observe that existing methods fail to maintain an optimal balance between stability and plasticity. Specifically, these methods often introduce additional constraints to mitigate catastrophic forgetting when learning new classes. For example, traditional regularization-base methods typically employ knowledge distillation loss (Li & Hoiem, 2017; Zhu et al., 2021) or regularization of parameters (Kirkpatrick et al., 2017); Recent LoRA-based methods (Liang & Li, 2024; Wang et al., 2023) leverage the idea of orthogonality to restrict the updating of new tasks, thereby reducing inter-task interference. However, when no information is leaked, excessive constraints can hinder the learning of new classes (e.g., InfLoRA), while focusing solely on new classes often leads to forgetting old tasks (e.g., L2P). This ultimately results in an imbalance between stability and plasticity for old and new tasks.

In this paper, we first discard the common paradigm of introducing constraints to mitigate catastrophic forgetting when learning new classes. Instead, The LoRAs are updated independently at each stage, allowing them to fully learn each task. At this point, the core question to address is *how to integrate knowledge from old and new tasks in order to balance stability and plasticity while improving overall accuracy.* To this end, we design two efficient post-processing strategies named **D**ynamic r**E**presentation con**S**olidation and dec**I**sion bounda**R**y r**E**finement (**DESIRE**). On the one hand, inspired by model fusion tasks that merge the backbones of models trained on different datasets during inference (Ilharco et al., 2022), we propose a *Continual Merging Paradigm* for LoRA-based continual learning. Specifically, we uniformly keep only two sets of previous and current LoRA parameters at each stage and merge them during inference. To better consolidate the representation of the merged model, a feature representation attribution loss is designed to learn the merging coefficients of the previous and current LoRA modules using a *tiny subset of unlabeled test data*. On the other hand, the independent training at different stages leads to classifiers that struggle to learn more generalized decision boundaries. To address this issue, we propose to refine the decision boundary of the classifier by reconstructing the high-dimensional feature distribution of the classes and sampling pseudo-features to retrain the classifier. Results on multiple datasets demonstrate the superiority of our method and a schematic of DESIRE is shown in Fig. 2.

In general, this paper makes three contributions: (i) We propose a new LoRA-based rehearsal-free CIL method that avoids introducing additional constraints when learning new classes to mitigate catastrophic forgetting, thereby maximizing the performance on each task. (ii) To better integrate knowledge from old and new tasks, we propose two efficient post-processing strategies, which can significantly improve performance by using only a tiny amount of unlabeled test data and statistical information from the training data. (iii) Experimental results indicate that our method significantly outperforms other rehearsal-free methods and performs comparable with latest rehearsal-based method (Lin et al., 2024) across multiple datasets and achieves the best balance between stability and plasticity.

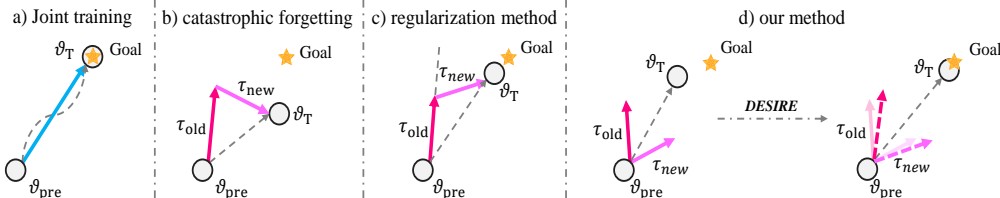

Figure 2: (a) Joint training using the full data achieves optimal performance (Upper bound). (b) Fine-tuning old models using only new data can lead to catastrophic forgetting. (c) Regularization-based methods protect old tasks by imposing additional constraints when learning new tasks. (d) Our method integrates knowledge by merging parameters from previous and current tasks and proposes DESIRE to consolidate the feature representation and refine the classifier.

## 2 RELATED WORK

**Parameter-Efficient Fine-Tuning.** To achieve better performance, training larger models is gradually becoming more mainstream (Achiam et al., 2023; Yin et al., 2023; Zhao et al., 2023). Although large models can cover multiple tasks, fine-tuning such a large model can become troublesome when addressing specific downstream tasks. To address this, Parameter-Efficient Fine-Tuning (PEFT) methods primarily based on LoRA (Hu et al., 2021), Prompt (Jia et al., 2022), and Adapter (Houlsby et al., 2019) have emerged. Specifically, LoRA reduces the number of parameters by parallelizing low-rank matrices at the attention layers of a frozen pre-trained model. Prompt tuning inserts additional tokens into the input embeddings at each layer and trains only these tokens during training. Adapter is similar to LoRA, but is usually serially embedded after specific layer of a pre-trained model. In summary, all of these methods fine-tune the entire large model with a small number of trainable parameters and can rival full fine-tuning in terms of performance (Fu et al., 2022; Hung et al., 2019; Zaken et al., 2021).

**Class Incremental Learning.** Existing CIL methods can be broadly categorized into expansion-based, regularization-based and rehearsal-based methods (Wang et al., 2024). With the widespread use of pre-trained models in recent years, expansion-based CIL methods using PEFT have gained significant attention. For instance, L2P (Wang et al., 2022c) maintains a prompt pool to select appropriate prompts for optimization across different tasks, and designs a frequency penalty loss to encourage diversified selection. LAE (Gao et al., 2023) introduces a CIL framework that is compatible with Adapter, Prompt and LoRA, demonstrating the extensibility of PEFT for CIL tasks. For LoRA-based methods, InfLoRA (Liang & Li, 2024) maintains a projection matrix to ensure that the LoRA parameters for the new task remain orthogonal to the inputs of the old task. O-LoRA (Wang et al., 2023), on the other hand, maintains a series of parameter matrices for the old tasks to constrain the model's updates to the parameter space orthogonal to the old tasks. However, the performance of these methods drops dramatically in the absence of information leakage and lacks a balance between stability and plasticity. Some recent works (Chitale et al., 2023; Guo et al., 2024) have also utilized the idea of merging old and new LoRA parameters for continual learning, but they all store the parameters from each stage and merge them with pre-defined coefficients. This paradigm will become redundant in long-term continual learning tasks (e.g., $T = 20$) because the model needs to store too many parameters from old tasks, while the fixed merging coefficient also limits performance. To this end, we propose a continual merging paradigm, where only the two parameter sets of the previous and current tasks are retained during fusion, and the merging coefficients are dynamically learned to better consolidate representation.

## 3 METHODOLOGY

### 3.1 PRELIMINARIES

**Class Incremental Learning:** CIL aims to learn a sequence of tasks $\{1, ..., T\}$, where each task $t$ contains a training dataset $\mathcal{D}_t = \{\mathbf{X}_t, \mathbf{Y}_t\} = \{x_j^t, y_j^t\}_{j=1}^{N_t}$ and $N_t$ denotes the number of training samples in the current task. The class sets between different tasks are disjoint. Formally, we define

the model to consist of two parts: a feature extractor $\mathcal{F}_\theta$ and a classifier $\mathcal{G}_\phi$. When learning task $t$, the loss function of CIL methods can usually be expressed as the following two parts:

$$\mathcal{L}(\theta^{(t)}, \phi^{(t)}) = \mathcal{L}_{ce}(\mathcal{G}(\mathcal{F}(\mathbf{X}_t; \theta_t); \phi_t), \mathbf{Y}_t) + \Omega_t, \tag{1}$$

where $\mathcal{L}_{ce}(\mathcal{G}(\mathcal{F}(\mathbf{X}_t; \theta_t); \phi_t), \mathbf{Y}_t)$ denotes the cross-entropy loss, and $\Omega_t$ represents the loss of regularization imposed in order not to forget old task knowledge. For example, $\Omega_t$ can be realized by knowledge distillation loss (Li & Hoiem, 2017; Zhu et al., 2021), parameter regularization loss (Wang et al., 2023) and so on. In addition to the constraints imposed by the loss function, Liang & Li (2024) designs the parameter subspace in advance of learning a new task, Other methods overcome catastrophic forgetting by embedding appropriate modules in reasoning, but also require training additional selection modules (Wang et al., 2022c; Yu et al., 2024).

**Low-Rank Adaption:** LoRA (Hu et al., 2021) assumes that updating to the parameters of the large language model during downstream task training lies on the low-rank space, and thus proposed to achieve comparable results to full fine-tuning by training only low-rank matrices concatenated in the original parameter space. Specifically, we define the linear layer in the pre-trained model as $\boldsymbol{W} \in \mathbb{R}^{d \times k}$, LoRA decomposes it into two low-rank matrices: $\boldsymbol{A} \in \mathbb{R}^{d \times r}$ and $\boldsymbol{B} \in \mathbb{R}^{r \times k}$, where $r \ll \min\{d, k\}$. By doing so, the forward propagation process in the linear layer can be re-expressed as $z = (\boldsymbol{W} + \boldsymbol{AB})x$, where $z$ and $x$ represent the outputs and inputs of the linear layer. In the implementation, in order not to affect the output of the model at the beginning, $\boldsymbol{A}$ is initialized by a random Gaussian, while $\boldsymbol{B}$ is initialized with zero. In our method, we insert LoRA at the $Q$ and $V$ matrices in the self-attention module at each block of the pre-trained transformer model. For clarity, we use LoRA inserted at $Q$ as an example in all subsequent discussions.

### 3.2 DYNAMIC REPRESENTATION CONSOLIDATION AND DECISION BOUNDARY REFINEMENT

Our method can be divided into the following three steps: individual training without additional constraints (Sec. 3.2.1), dynamic representation consolidation (Sec. 3.2.2), and decision boundary refinement (Sec. 3.2.3). Fig. 3 illustrates the framework of our method.

#### 3.2.1 INDIVIDUAL TRAINING WITHOUT CONSTRAINTS

Unlike existing methods that require additional constraints to protect information from old tasks while training the current task, we treat each training stage as independent of the others. Specifically, the LoRA parameters are reinitialized at each stage and only the cross-entropy loss $\mathcal{L}_{ce}$ in Eq. (1) is optimized. This has two benefits: (i) Individual training allows the model to focus on improving the performance of the current task, which indirectly enhances performance after merging. (ii) We find that since the LoRA is reinitialized for each task, it naturally maintains good orthogonality with the parameter space of previous tasks after training (See Appendix A.3), which creates a solid prerequisite for the fusion of model parameters.

In order to perform the dynamic representation consolidation and decision boundary refinement, we count the statistical information of each class after training each task. Specifically, we assume that the features learned by the feature extractor can be approximated by a mixture of Gaussian distributions (Luo et al., 2021; Lindsay, 1995). Therefore, the feature distribution of class $i$ can be reconstructed by counting the mean $\boldsymbol{\mu}_i$ and covariance $\boldsymbol{\Sigma}_i$ matrices:

$$\boldsymbol{\mu}_i = \frac{1}{N_i} \sum_{j=1}^{N_i} z_{i,j}, \quad \boldsymbol{\Sigma}_i = \frac{1}{N_i - 1} \sum_{j=1}^{N_i} (z_{i,j} - \boldsymbol{\mu}_i)(z_{i,j} - \boldsymbol{\mu}_i)^T, \tag{2}$$

where $z_{i,j} = \mathcal{F}(\boldsymbol{x}_{i,j}; \theta_t)$ is the feature of the $j$-th sample and $N_i$ is the number of training data of class $i$. In the implementation, we keep $\boldsymbol{\mu}_i$ and $\boldsymbol{\Sigma}_i$ for all the classes the model has seen.

#### 3.2.2 PARAMETERS MERGING WITH DYNAMIC REPRESENTATION CONSOLIDATION

By individually training, we obtain the LoRA parameters for each task and denote them by $\{\theta_t^{1,\boldsymbol{A}}, ..., \theta_t^{l,\boldsymbol{A}}\}$ and $\{\theta_t^{1,\boldsymbol{B}}, ..., \theta_t^{l,\boldsymbol{B}}\}$, where $t$ represents the task identity and $l$ denotes the number of blocks in pre-trained model. In practice, new tasks would emerged continually, and it is crucial to get a unified model that embraces all the task information through these individual model

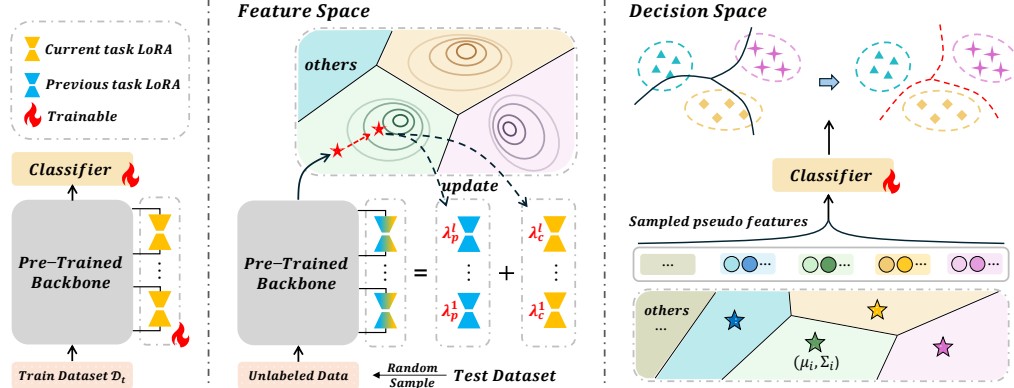

Figure 3: Illustration of the proposed DESIRE. **Left:** The backbone of the model is frozen during individual training and only the LoRA and classifier are trainable. **Middle:** We obtain the knowledge of the old and new tasks by merging the parameter space (LoRA). To better consolidate the representations, we sample *tiny unlabeled test data* to optimize the merging coefficients through our proposed attribution loss (Sec. 3.2.2). **Right:** We reconstruct the pseudo-features using the counted statistical informations and use them to refine the decision boundaries of the classifier.

parameters. Existing model merging researches (Ilharco et al., 2022; Chitale et al., 2023) have shown that fusing different tasks directly on the parameter space is promising. The merging process can be expressed formally as $\theta_m = \theta_{init} + \sum_{t=1}^{T} \lambda_t \tau_t$, where $\tau_t = \theta_t - \theta_{init}$ and $\lambda_t$ is a scaling hyperparamter. Unlike these model merging methods that focus on merging at the model backbone level, we concentrate on acquiring knowledge of old and new tasks through merging LoRAs. In the CIL community, some recent works (Chitale et al., 2023; Sun et al., 2023; Zheng et al., 2023) have also proposed to overcome catastrophic forgetting through model merging. However, these methods inherit the model merging paradigm directly, which retains the parameters of each stage and assigns an empirical coefficient (e.g., $1/T$) for direct merging. Although good results can be achieved, it is inappropriate for the CIL task. Specifically, on the one hand, CIL learns a much larger number of tasks (e.g., $T = 20$), and *it is not practical to store the parameters of all previous tasks in each subsequent stage*. On the other hand, the choice of the merging hyperparameters could have a significant impact on the performance as it directly affects the feature representation of the merged model, while these existing methods usually use fixed empirical values. To this end, we propose a new *continual merging paradigm* to better address these two issues.

**Continual Merging Paradigm.** To avoid storing LoRA parameters for each task, we define the model to keep only two sets of parameters for the *current* and *previous* at each task $t$ (e.g., $\{\theta_{t,c}^{1,\boldsymbol{A}}, ..., \theta_{t,c}^{l,\boldsymbol{A}}\}$ and $\{\theta_{t,p}^{1,\boldsymbol{A}}, ..., \theta_{t,p}^{l,\boldsymbol{A}}\}$). Specifically, we leverage greedy algorithm to obtain the *previous* parameters:

$$\theta_{t,p}^{i,\boldsymbol{A}} = \lambda_{t-1,c}^{i,\boldsymbol{A}} * \theta_{t-1,c}^{i,\boldsymbol{A}} + \lambda_{t-1,p}^{i,\boldsymbol{A}} * \theta_{t-1,p}^{i,\boldsymbol{A}}, \tag{3}$$

where $\lambda_{t-1,c}^{i,\boldsymbol{A}}$ and $\lambda_{t-1,p}^{i,\boldsymbol{A}}$ represent the learned merging coefficients of block $i$ obtained from $t-1$ task . The same merging operation is applied to the $\boldsymbol{B}$ matrix as well. This not only integrates all the old task parameters into one set, which greatly reduces the memory occupy (from $(T-1) \cdot l$ to $l$), but also avoids reinitializing the merging coefficients of all tasks at the time of consolidation, which improves the convergence speed. We compare different merging methods in Sec 4.3 to better emphasize the superiority of our paradigm.

To better consolidate the feature representation of the merged model, we propose to learn the merging coefficients by minimising the entropy of the model output distribution (Grandvalet & Bengio, 2004; Roy et al., 2022). However, in CIL tasks, directly optimizing entropy minimisation loss using logits from the classifier's output is not reasonable, as the classifier tend to be baised towards newly learned classes (Wu et al., 2019; Hou et al., 2019). To this end, we propose a *feature-level attribution loss* to update the merging coefficients using the mean $\boldsymbol{\mu}_i$ and covariance $\boldsymbol{\Sigma}_i$ counted after individual training (Sec 3.2.1). Specifically, with the $\boldsymbol{\mu}_i$ and $\boldsymbol{\Sigma}_i$, the distribution of the class $i$ in the feature space can be reconstructed. For a feature representation $\boldsymbol{z}_j$ of a test sample $\boldsymbol{x}_j$, the logarithm

of the probability density between $z_j$ and the feature distribution of each class is:

$$\boldsymbol{\sigma}^i = \log \varphi(\boldsymbol{z}; \boldsymbol{\mu}_i, \boldsymbol{\Sigma}_i)$$
$$= -\frac{1}{2}[(z_j - \boldsymbol{\mu}_i)^T \boldsymbol{\Sigma}_i^{-1}(z_j - \boldsymbol{\mu}_i) + d\log(2\pi) + \log|\boldsymbol{\Sigma}_i|], \tag{4}$$

where $d$ is the dimension of the feature. Intuitively, $\boldsymbol{\sigma}^i$ represents the similarity between $z_j$ and the distribution of class $i$. Therefore, $\boldsymbol{\sigma}$ can be served as a surrogate logits for entropy minimization loss. In summary, the proxy optimization objective for the parameter space re-calibration phase can be expressed as:

$$\min_{\lambda_1,...,\lambda_N} \sum_{n=1}^{N} \sum_{x_j \in \mathcal{D}_m} \mathcal{H}(\hat{\boldsymbol{\sigma}}/\kappa), \text{ where } \hat{\boldsymbol{\sigma}} = \frac{\boldsymbol{\sigma} - \min(\boldsymbol{\sigma})}{\max(\boldsymbol{\sigma}) - \min(\boldsymbol{\sigma})}, \tag{5}$$

where $N$ is the number of LoRA parameters to be merged, $\mathcal{D}_m$ represents the sampled mini test dataset, $\mathcal{H}(\cdot)$ is the Shannon Entropy (Shannon, 1948) and $\kappa$ denotes the temperature coefficient (we set $\kappa = 0.1$ in our experiments).

### 3.2.3 DECISION BOUNDARY REFINEMENT

In addition to catastrophic forgetting in feature representation, confusion of decision boundaries at the classifier level also limits model performance. Existing literature suggests that training a model without data from old tasks makes the classifier heavily biased towards newly learned classes (Wu et al., 2019; Hou et al., 2019), leading the model to misclassify old classes as new ones, thereby exacerbating the forgetting of old classes. To address this issue, we propose to refine the decision boundary by leveraging the sampled pseudo-features to calibrate the biased classifier. Specifically, with the statistical information $(\boldsymbol{\mu_i}, \boldsymbol{\Sigma_i})$ of each class $i$ obtained from Eq 2, we can reconstruct the feature distribution $\mathcal{N}_i$, and the pseudo-features $\hat{\mathcal{Z}}_i = \{\hat{z}_{i,1}, ..., \hat{z}_{i,N_i}\}$ of class $i$ can be formed by sampling from the distribution $\mathcal{N}_i$, where $N_i$ is the number of pseudo-features for each class. Then, we optimize the classifier with the set of pseudo-features of all seen classes $\hat{\mathcal{Z}} = [\hat{\mathcal{Z}}_1, ..., \hat{\mathcal{Z}}_C]$ directly through cross-entropy loss:

---

**Algorithm 1** Our proposed DESIRE

**Inputs:** Pre-trained backbone $\mathcal{F}_\theta$; classifier $\mathcal{G}_\phi$; *current* LoRA $\theta_c$ and *previous* LoRA $\theta_p$; training dataset $\mathcal{D}_t$ and mini merging dataset $\mathcal{D}_m$.
**Output:** Unified model $\mathcal{F}_{\hat{\theta}} \propto \mathcal{G}_{\hat{\phi}}$.

1: **for** $t = 1 \to T$ **do**
2:     # individual training.
3:     **for** $e = 1 \to Epoch_{ind}$ **do**
4:         Train $\theta_c$ and $\mathcal{G}_\phi$ with $\mathcal{L}_{ce}$ in Eq. (1) on $\mathcal{D}_t$.
5:     **end for**
6:     Calculate $\boldsymbol{\mu}_i$ and $\boldsymbol{\Sigma}_i$ using Eq. (2)
7:     # Dynamic representation consolidation.
8:     **for** $e = 1 \to Epoch_r$ **do**
9:         Train merging coefficients $\lambda_c$ and $\lambda_p$ with $\boldsymbol{\mu}$ and $\boldsymbol{\Sigma}$ using Eq. (5) on $\mathcal{D}_m$.
10:    **end for**
11:    Consolidate $\theta_p$ with $\lambda_c$ and $\lambda_p$ using Eq. (3).
12:    # Decision boundary refinement.
13:    Sample $\hat{\mathcal{Z}}$ from $\mathcal{N}(\boldsymbol{\mu}, \boldsymbol{\Sigma})$ for all seen classes.
14:    **for** $e = 1 \to Epoch_c$ **do**
15:       Train classifier $\mathcal{G}_{\hat{\phi}}$ with pseudo-features $\hat{\mathcal{Z}}$ using Eq. (6).
16:    **end for**
17: **end for**

---

$$\min_{\phi} \sum_{i=1}^{C} \sum_{j=1}^{N_i} \mathcal{L}_{ce}(\mathcal{G}_\phi(\hat{z}_{i,j}), y_i), \tag{6}$$

where $C$ is the number of all seen classes. We thus obtain the calibrated feature extractors $\mathcal{F}_{\hat{\theta}}$ and classifier $\mathcal{G}_{\hat{\phi}}$ that are used for subsequent inference. It is worth mentioning that our post-processing module requires only a minimal amount of training time at the end of each stage (See Sec 4.4).

**Remarks.** Although both our method and SLCA (Zhang et al., 2023) enhance the classifier by sampling features for retraining, our method is superior in reconstructing the feature distribution. This is primarily because SLCA trains the entire backbone sequentially, causing the old feature space to inevitably drift when training new classes, even if the learning rate is low. In contrast, our method effectively maintains the independence of each task's feature space by training them individually and combines them by calibrating the parameter space, ensuring consistency between the reconstructed distribution and the true distribution. A detailed analysis is provided in Appendix A.4.

Table 1: Comparison of the performance of different CIL methods. The best result in each setting is highlighted in **bold**. We report the upper bound on performance under each setting in **Joint**, which is obtained by training each task with the dataset of all seen classes. The rehearsal-free methods (**below**) and rehearsal-based methods (**above**) are divided into two parts by the dashed line. Our methods belongs to the rehearsal-free method and the results in table are marked in gray. The detailed results with standard deviation can be seen in Tables 4 - 6 of Appendix A.2.

| | C100-5T | | C100-10T | | T200-5T | | T200-10T | | I380-5T | | I380-10T | |
| | $A_{last}$ | Avg | $A_{last}$ | Avg | $A_{last}$ | Avg | $A_{last}$ | Avg | $A_{last}$ | Avg | $A_{last}$ | Avg |
|---|---|---|---|---|---|---|---|---|---|---|---|---|
| **Joint** | 81.83 | 87.24 | 81.51 | 88.05 | 71.64 | 77.43 | 71.54 | 78.28 | 79.50 | 84.01 | 79.87 | 84.73 |
| MEMO | 64.36 | 77.36 | 60.12 | 75.60 | 43.28 | 61.72 | 35.74 | 58.35 | 51.22 | 66.40 | 49.79 | 67.98 |
| FOSTER | 71.36 | 81.23 | 70.89 | 81.08 | 57.58 | 70.09 | 56.09 | 68.80 | 62.93 | 74.65 | 64.29 | 74.15 |
| MORE | 71.38 | 80.64 | 69.82 | 79.78 | 62.79 | 71.94 | 60.44 | 70.88 | 72.75 | 80.63 | 69.26 | 78.44 |
| ROW | 74.96 | 83.11 | 74.01 | 83.31 | 62.79 | 71.72 | 61.76 | 72.29 | 73.12 | 80.63 | 72.09 | 80.61 |
| TPL | **75.87** | **84.11** | **75.02** | **84.67** | **66.86** | **75.04** | **64.89** | **74.67** | **76.88** | **82.69** | **75.32** | **81.79** |
| LAE-Adapter | 68.72 | 78.58 | 66.01 | 77.45 | 63.32 | 72.48 | 60.04 | 70.71 | 66.13 | 76.02 | 59.85 | 71.46 |
| LAE-Prefix | 68.52 | 78.67 | 65.73 | 77.09 | 63.12 | 72.29 | 58.99 | 69.93 | 70.02 | 78.67 | 64.55 | 75.14 |
| LAE-LoRA | 68.66 | 78.91 | 65.75 | 77.75 | 63.58 | 72.63 | 59.57 | 70.69 | 69.73 | 78.15 | 64.49 | 75.75 |
| L2P | 67.73 | 78.47 | 63.26 | 75.47 | 60.91 | 70.03 | 56.39 | 68.47 | 66.22 | 74.70 | 62.41 | 71.14 |
| DualPrompt | 68.08 | 78.22 | 63.83 | 75.25 | 60.44 | 69.53 | 57.53 | 68.65 | 65.54 | 75.58 | 62.86 | 74.40 |
| CODA-Prompt | 70.36 | 80.22 | 66.28 | 77.85 | 61.98 | 71.42 | 58.44 | 69.91 | 68.93 | 77.65 | 65.04 | 75.82 |
| PASS | 72.19 | 81.20 | 68.97 | 79.36 | 63.33 | 72.44 | 60.62 | 71.20 | 67.49 | 76.09 | 64.40 | 74.88 |
| O-LoRA | 67.32 | 78.16 | 64.35 | 77.16 | 61.45 | 72.22 | 60.66 | 71.59 | 59.95 | 75.98 | 58.28 | 72.09 |
| EASE | 68.72 | 77.66 | 66.15 | 77.49 | 56.93 | 66.36 | 56.70 | 67.70 | 64.88 | 72.71 | 64.40 | 73.78 |
| InfLoRA | 69.66 | 79.70 | 63.86 | 76.31 | 56.43 | 68.36 | 56.43 | 68.36 | 72.50 | 80.30 | 67.53 | 77.57 |
| **Ours** | 72.89 | 81.34 | 72.47 | 81.55 | 64.42 | 73.68 | 64.36 | 74.56 | 74.90 | 81.83 | 72.69 | 81.29 |

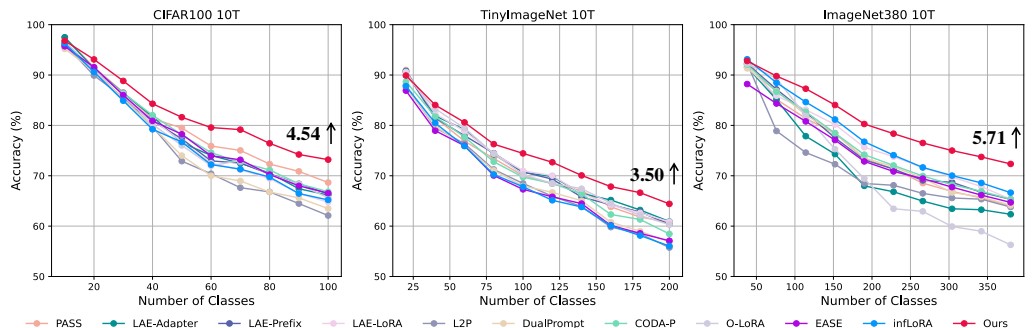

Figure 4: Results of accuracy curve on CIFAR100, TinyImageNet and ImageNet380 under 10T.

# 4 EXPERIMENTS

## 4.1 EXPERIMENTS SETUP

**Baselines.** We compare our methods with state-of-the-art continual learning methods, including *rehearsal-free methods*: PASS (Zhu et al., 2021), LAE (Gao et al., 2023), L2P (Wang et al., 2022c), DualPrompt (Wang et al., 2022b), CODA-Prompt (Smith et al., 2023), O-LoRA (Wang et al., 2023), EASE (Zhou et al., 2024), InfLoRA (Liang & Li, 2024), and *rehearsal-based methods*: iCaRL (Rebuffi et al., 2017), DER (Yan et al., 2021), FOSTER (Wang et al., 2022a), MORE (Kim et al., 2022), ROW (Kim et al., 2023) and TPL (Lin et al., 2024). For fair comparison, we re-ran the corresponding open-source codes for each method using the same pre-trained weights. For methods not based on PEFT, we freeze the backbone and fine-tune it with LoRA.

**Architecture and Training Details.** In order to exclude information leakage due to the class overlap between the data used in the pre-trained models and the experiment data, we follow the setup in Lin et al. (2024); Kim et al. (2023; 2022) and use the same Deit-S/16 model (Touvron et al., 2021), which uses the ImageNet-1k (Russakovsky et al., 2015) in removing the classes that are similar or identical to the CIFAR100 (Krizhevsky et al., 2009) and TinyImageNet (Le & Yang, 2015) for pre-training. The LoRAs are inserted in the query and value of the self-attention module and freeze the model backbone during training to train only LoRA modules and classifier. In experiments, the rank of LoRA is set to 4, and the number of epochs for dynamic representation consolidation and decision

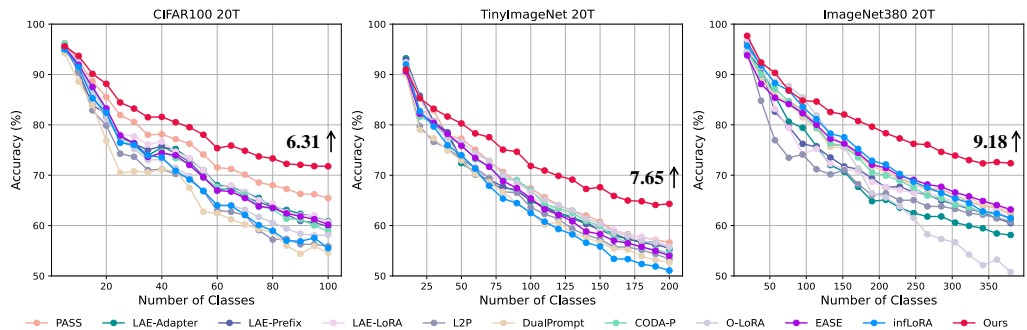

Figure 5: Results of accuracy curve on CIFAR100, TinyImageNet and ImageNet380 under 20T.

boundary refinement are set to 5 and 10, respectively. The size of *tiny unlabeled test data* are set to 500, 1000 and 1900, respectively. **More implementation details** are given in Appendix A.1.

**Datasets and Evaluation Metric.** We use CIFAR100, TinyImageNet and ImageNet380 (Lin et al., 2024) to train and evaluate the models. CIFAR100 and TinyImageNet are widely used in exist CIL works. Imagenet380 is a large-scale dataset consisting of 380 classes randomly selected from the 389 classes removed from ImageNet-1k. Following existing CIL work (Wang et al., 2022c), we first randomly shuffle the class order of the datasets and then split then into 5, 10 and 20 tasks. We report the averaged metrics over 3 random orders. For *rehearsal-based methods*, the replay buffer size is set as 1000 for CIFAR100 and TinyImageNet, and 3800 for ImageNet380. Note that our DESIRE is *rehearsal-free method* and we do not save any old samples.

We report the standard metrics to evaluate the CIL methods (Lin et al., 2024): $A_{last}$ is computed as the accuracy of all seen classes that have already been learned after learning the final task. ***Avg*** is computed as the average accuracy of each task: $\boldsymbol{Avg} = \frac{1}{T} \sum_{t=1}^{T} A_t$, where $T$ is the total number of tasks and $A_t$ is the accuracy of all seen classes that have learned after learning task $t$.

## 4.2 EXPERIMENTAL RESULTS

A summary of the results is provided in Table 1, Fig. 4 and 5. The detailed results with standard deviation can be seen in Tables 4 - 6 and Fig. 9 of Appendix A.2. Our method outperforms existing rehearsal-free methods across three datasets (CIFAR100: **C100**, TinyImageNet: **T200** and ImageNet380: **I380**) and three task settings (**5T**, **10T** and **20T**), achieving an average improvement of **4.42%** and **3.08%** on $A_{last}$ and ***Avg*** metrics, respectively. It is noteworthy that when there is no information leakage, existing PEFT-based methods all show varying degrees of degradation, while PASS, which is not designed for PEFT, achieves relatively higher performance. This result aligns with the findings of TPL (Lin et al., 2024). However, PASS suffers from high training time overhead, whereas our method significantly enhances the performance of rehearsal-free method in a limited amount of time (See Sec 4.4). Meanwhile, we observe that for the same dataset, all other methods experience a significant decrease in the $A_{last}$ metric as the number of tasks increases (e.g., InfLoRA drops from **72.50%** to **61.80%** on ImageNet380). This suggests that when the number of tasks is small (e.g., $T = 5$), existing methods can effectively overcome catastrophic forgetting by imposing additional constraints. However, as the number of tasks increases, the constraints imposed to protect old tasks will continuously squeeze the solution space for new tasks, resulting in a significant decrease in performance. In contrast, our method maintains the solution space for each task as much as possible and organically combines old and new tasks through dual calibration, thereby greatly reducing the performance difference between the number of short and long tasks (our method drops from **74.90%** to **72.45%** on ImageNet380). Compared to rehearsal-based methods, our method does not impose the stringent requirement of preserving old task samples and significantly reduces the gap with latest rehearsal-based methods.

## 4.3 ABLATION STUDY

**Performance results.** Fig. 6 (a) illustrates the effect of each compoent on the $A_{last}$ and ***Avg*** metrics. We use the traditional model merging paradigm as a baseline, with the merging coefficients set

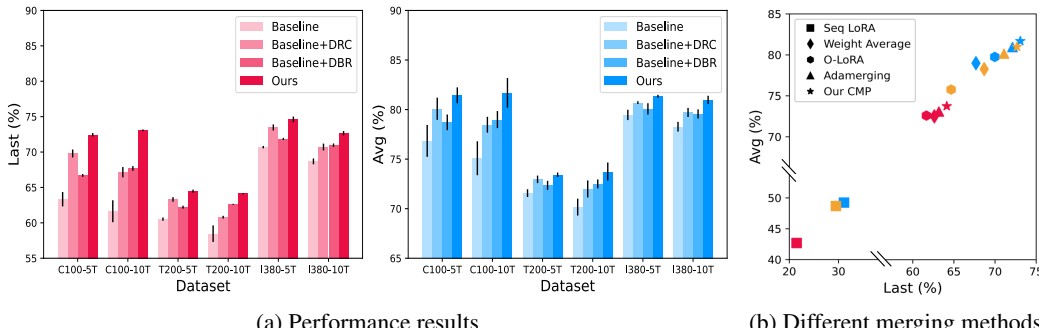

(a) Performance results         (b) Different merging methods

Figure 6: Ablation Studies. (a) Performance gains on three datasets by adding each component. (b) Performance comparison of different merging methods on C100-10T, T200-10T and I380-10T.

to the empirical values ($\lambda = 1/T$). The average $A_{last}$ and *Avg* metrics are 63.89% and 75.23%, respectively. After consolidating the representation (Baseline+DRC), the performance improves by **3.65%** and **2.09%**, respectively. This suggests that dynamically update the merging coefficients can provide better feature representations and thus improve performance. Decision boundary refinement (Baseline+DBR) can yield a **3.11%** and **1.80%** performance improvement, demonstrating that classifiers obtained through direct concatenation suffer from decision boundary confusion, while our post-calibration can effectively modified the classifiers. When the two modules are added, they can be organically combined and bring about a **6.40%** and **3.81%** performance improvement.

**Different Merging Methods.** We compare various merging strategies for LoRA based on decision space calibration and the results are presented in Fig. 6 (b). *Seq LoRA* refers to sequential fine-tuning of the same LoRA, which inevitably leads to catastrophic forgetting. *Weight average* refers to saving the LoRA parameters for each task and merging all LoRA parameters on average at the $t$-th task during inference. It is evident that direct average merging can effectively mitigate catastrophic forgetting and we provide further analysis in Appendix A.3. *O-LoRA* (Wang et al., 2023) acquires knowledge of old and new tasks by concatenating LoRA instead of merging and introduces an orthogonality loss in the parameter space. However, enforcing strict orthogonality in the parameter space may hinder the model's ability to learn general information across tasks, thereby limiting performance improvements. *Adamerging* (Yang et al., 2023) updates the merging coefficients by optimizing the entropy-minimizing loss of the logits obtained from the classifier, but the logits obtained by the classifier are suboptimal. Our proposed *continual merging paradigm* (CMP) avoids error accumulation due to classifier drift by computing the attribution degree in the feature space instead of logit. Moreover, we retain only two sets of LoRA parameters at each stage instead of saving all the LoRA parameters, which significantly reduces memory usage.

## 4.4 FURTHER ANALYSIS

**Stability and plasticity analysis.** We emphasize that a robust CIL method requires not only high performance on both $A_{last}$ and *Avg* metrics, but also a balance between stability and plasticity, especially for long-phase tasks. In the previous section, we primarily highlighted the results of our method on the first two metrics, and in this section we focus on analyzing the stability and plasticity of different methods. In Fig. 1, we visualize the accuracy of current task and average accuracy of all previous

Table 2: Results of SD(Acc) on three datasets under long-phase settings ($T = 10$ and 20).

| | CIFAR100 | | TinyImageNet | | ImageNet380 | | Average |
|---|---|---|---|---|---|---|---|
| | 10T | 20T | 10T | 20T | 10T | 20T | |
| LAE-Adapter | 9.22 | 11.80 | 7.26 | 6.34 | 20.28 | 17.16 | 12.01 |
| LAE-Prefix | 6.71 | 10.06 | 8.95 | 7.64 | 9.94 | 10.15 | 8.91 |
| LAE-LoRA | 8.39 | 10.35 | 7.40 | 7.00 | 6.21 | 7.98 | 7.89 |
| PASS | 4.18 | 5.66 | 3.48 | 5.30 | **3.14** | **3.50** | 4.21 |
| L2P | 8.23 | 11.07 | 7.78 | 8.73 | 17.87 | 17.64 | 11.89 |
| DualPrompt | 7.34 | 11.58 | 6.98 | 9.79 | 11.31 | 6.18 | 8.86 |
| CODA-Prompt | 5.41 | 10.44 | 7.41 | 9.22 | 5.84 | 6.82 | 7.52 |
| O-LoRA | 5.18 | 11.17 | 4.39 | 6.96 | 4.23 | 5.50 | 6.24 |
| EASE | 5.81 | 10.84 | 6.16 | 8.20 | 6.45 | 7.19 | 7.45 |
| InfLoRA | 5.19 | 12.13 | 4.33 | 9.15 | 3.37 | 4.93 | 6.52 |
| **Ours** | **4.10** | **5.59** | **3.06** | **5.00** | 3.25 | 3.65 | **4.11** |

tasks in the last stage. It can be observed that existing methods experience dvarying degrees of imbalance across three datasets, and only our method achieves the highest robustness. Additionally, we define *Standard Deviation of task-wise accuracy (SD(Acc))* for quantitative analysis, which cal-

culates the standard deviation of the model's accuracy in the last stage for each task. Specifically, a lower *SD(Acc)* indicates a less fluctuation and more robust performance across different tasks. In contrast, a higher SD(Acc) indicates that performance variations across different tasks are more pronounced, making it more likely to forget certain tasks. We present the results for long-phase settings ($T = 10$ and 20) across three datasets. As shown in Table 2, our method achieves the best performance among all methods, while the other methods fluctuate significantly at different settings. It is noteworthy that PASS also achieves similar results to ours. However, when considering both $\boldsymbol{A}_{last}$ and *Avg* metrics, our method significantly outperforms all other methods.

**Merging coefficients analysis**. Fig. 7 illustrates the merging coefficients learned by our proposed *continual merging paradigm* under CIFAR100-5T, where $Q_p$ and $Q_c$ represents the LoRAs previously and currently embedded in $Q$ matrix of the self-attention. $\boldsymbol{A}_i$ and $\boldsymbol{B}_i$ represent the downsampling and upsampling matrices of the LoRA for layer $i$. We observe that: (i) The merging coefficients generally remain stable, which in conjunction with the analyses in Appendix A.3 illustrates that model parameter merging can effectively integrate knowledge from old and new tasks. (ii) Only a small amount of variation in the merging coefficients plays a crucial role in performance improvement, and these variations can be efficiently captured through training. (iii) The changes in the merging coefficients of the LoRAs embedded in $Q$ and $V$ exhibit different trends: the former updates focus on the top layer, while the latter concentrate near the middle layer. This suggests that LoRAs at different locations capture distinct information and simple average merging could lose this specificity, potentially leading to sub-optimal performance.



Figure 7: Visualisation of merging coefficients.

**Training efficiency analysis**. For CIL tasks, especially those based on pre-trained models, models are expected to quickly adapt and acquire knowledge of both old and new tasks, thereby accelerating their application to downstream tasks. We measure the average time taken to train *each epoch* for different methods on the same device (RTX 4090). For a fair comparison, we set the batch size to 64 for both. As can be seen in Fig. 8, although our method consists of three stages, it does not require excessive additional training time. In contrast, the self-supervised learning and knowledge distillation strategies employed in PASS take up a significant amount of training time, which contradicts the purpose of rapidly adapt-

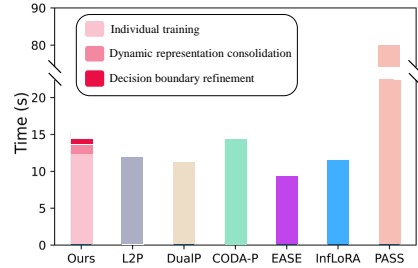

Figure 8: Demonstration of training efficiency.

ing to downstream tasks using the PEFT-based CIL method. Moreover, our method achieves significantly better performance than other rehearsal-free methods while adapting rapidly.

## 5 CONCLUSION

In this paper, we propose a novel PEFT-based rehearsal-free CIL method named **DESIRE**. Our method fully learns each task by training each stage independently and integrates knowledge from both old and new tasks through efficient dynamic representation consolidation and decision boundary refinement to overcome catastrophic forgetting and improve model performance. Experimental results demonstrate that our method achieves state-of-the-art performance compared to the existing rehearsal-free methods, while maintaining a good balance between stability and plasticity.

**Limitations and future works:** While our proposed method demonstrates strong performance in image classification tasks, this represents only a subset of the broader potential of AI systems. In future work, we plan to extend and adapt our approach to tackle more complex and diverse visual tasks, such as object detection and image segmentation. Expanding to these areas will help us understand the method's broader applicability in real-world scenarios.

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

# A  APPENDIX

## A.1  IMPLEMENTATION DETAILS

The training configuration of our method on three datasets is shown in Table 3. For a fair comparison, we re-run the open-source code of other methods using the same pre-trained model and tune the performance of other methods as much as possible using our training configuration as a reference.

Table 3: Training configuration and hyperparameter settings

| | config | CIFAR100 | TinyImageNet | ImageNet380 |
|---|---|---|---|---|
| **Individual training (Sec 3.2.1)** | training epoch | 20 | 20 | 10 |
| | optimizer | SGD | SGD | SGD |
| | learning rate | 5e-3 | 5e-3 | 5e-3 |
| | momentum | 0.9 | 0.9 | 0.9 |
| | batch size | 64 | 64 | 128 |
| | scheduler | CosineAnnealing | CosineAnnealing | CosineAnnealing |
| **Dynamic representation consolidation(Sec 3.2.2)** | training epoch | 5 | 5 | 5 |
| | optimizer | SGD | SGD | SGD |
| | learning rate | 0.1 | 0.15 | 0.04 |
| | momentum | 0.9 | 0.9 | 0.9 |
| | batch size | 64 | 64 | 64 |
| | merge dataset size | 500 | 1000 | 1900 |
| | $\kappa$ in Eq.(3) | 0.1 | 0.1 | 0.1 |
| | $[\lambda_{p,init}, \lambda_{c,init}]$ | [0.5, 0.5] | [0.5, 0.5] | [0.5, 0.5] |
| **Decision boundary refinement(Sec 3.2.3)** | training epoch | 10 | 10 | 10 |
| | optimizer | SGD | SGD | SGD |
| | learning rate | 5e-3 | 5e-3 | 5e-4 |
| | momentum | 0.9 | 0.9 | 0.9 |
| | batch size | 64 | 64 | 64 |
| | $N_i$ | 200 | 200 | 200 |

## A.2 MAIN RESULTS

We report the quantitative results of different methods on three datasets (**CIFAR100**, **TinyImageNet** and **ImageNet380**) under three settings (**5T**, **10T** and **20T**) in Tables 4, 5 and 6. In Fig. 9. we also plot the performance curve of the different methods for different settings. Compared with the rehearsal-free methods, our method achieve an average improvement of **4.42%** and **3.08%** on $A_{last}$ and *Avg* metrics.

Table 4: Comparison of the performance of different CIL methods on CIFAR100. We test the results of different methods under *three class orders* and report the mean and standard deviation.

| | C100-5T | | C100-10T | | C100-20T | | Average | |
| --- | --- | --- | --- | --- | --- | --- | --- | --- |
| | $A_{last}$ | Avg | $A_{last}$ | Avg | $A_{last}$ | Avg | $A_{last}$ | Avg |
| **Joint** | $81.83^{\pm0.26}$ | $87.24^{\pm0.73}$ | $81.51^{\pm0.22}$ | $88.05^{\pm0.68}$ | $81.81^{\pm0.21}$ | $88.58^{\pm0.76}$ | 81.72 | 87.96 |
| MEMO | $64.36^{\pm0.25}$ | $77.36^{\pm0.90}$ | $60.12^{\pm0.50}$ | $75.60^{\pm0.55}$ | $53.29^{\pm1.91}$ | $71.78^{\pm1.14}$ | 59.26 | 74.91 |
| FOSTER | $71.36^{\pm0.50}$ | $81.23^{\pm0.39}$ | $70.89^{\pm0.52}$ | $81.08^{\pm0.68}$ | $69.54^{\pm0.26}$ | $80.16^{\pm0.78}$ | 70.60 | 80.82 |
| MORE | $71.38^{\pm1.01}$ | $80.64^{\pm0.98}$ | $69.82^{\pm0.41}$ | $79.78^{\pm0.96}$ | $67.92^{\pm1.08}$ | $78.35^{\pm1.64}$ | 69.71 | 79.59 |
| ROW | $74.96^{\pm0.20}$ | $83.11^{\pm0.62}$ | $74.01^{\pm0.21}$ | $83.31^{\pm0.91}$ | $74.29^{\pm0.23}$ | $83.92^{\pm0.67}$ | 74.42 | 83.45 |
| TPL | $\mathbf{75.87^{\pm0.26}}$ | $\mathbf{84.11^{\pm0.54}}$ | $\mathbf{75.02^{\pm0.19}}$ | $\mathbf{84.67^{\pm0.53}}$ | $\mathbf{74.98^{\pm0.47}}$ | $\mathbf{84.52^{\pm0.75}}$ | **75.29** | **84.43** |
| LAE-Adapter | $68.52^{\pm0.60}$ | $78.67^{\pm0.71}$ | $66.01^{\pm0.30}$ | $77.45^{\pm0.77}$ | $60.55^{\pm1.23}$ | $73.74^{\pm1.33}$ | 65.03 | 76.62 |
| LAE-Prefix | $68.72^{\pm0.78}$ | $78.58^{\pm0.58}$ | $65.73^{\pm0.68}$ | $77.09^{\pm0.89}$ | $59.92^{\pm0.66}$ | $72.68^{\pm0.56}$ | 64.79 | 76.12 |
| LAE-LoRA | $68.66^{\pm0.60}$ | $78.91^{\pm0.70}$ | $65.75^{\pm1.19}$ | $77.75^{\pm0.32}$ | $60.40^{\pm0.60}$ | $73.05^{\pm1.90}$ | 64.93 | 76.57 |
| L2P | $67.73^{\pm0.95}$ | $78.47^{\pm0.73}$ | $63.26^{\pm1.19}$ | $75.47^{\pm0.51}$ | $55.39^{\pm0.53}$ | $69.41^{\pm1.35}$ | 62.13 | 74.45 |
| DualPrompt | $68.08^{\pm0.42}$ | $78.22^{\pm0.59}$ | $63.83^{\pm0.42}$ | $75.25^{\pm0.64}$ | $55.36^{\pm1.87}$ | $69.37^{\pm2.08}$ | 62.42 | 74.28 |
| CODA-Prompt | $70.36^{\pm0.87}$ | $80.22^{\pm1.01}$ | $66.28^{\pm0.52}$ | $77.85^{\pm1.17}$ | $59.94^{\pm1.16}$ | $73.74^{\pm1.70}$ | 65.53 | 77.27 |
| PASS | $72.19^{\pm0.33}$ | $81.20^{\pm0.73}$ | $68.97^{\pm0.50}$ | $79.36^{\pm0.52}$ | $64.65^{\pm2.17}$ | $76.25^{\pm1.55}$ | 68.60 | 78.94 |
| O-LoRA | $67.32^{\pm0.95}$ | $78.16^{\pm1.12}$ | $64.35^{\pm1.26}$ | $77.16^{\pm2.0}$ | $58.59^{\pm0.54}$ | $72.08^{\pm1.64}$ | 63.42 | 75.80 |
| EASE | $68.72^{\pm0.22}$ | $77.66^{\pm0.82}$ | $66.15^{\pm0.42}$ | $77.49^{\pm1.36}$ | $60.04^{\pm0.16}$ | $73.66^{\pm1.34}$ | 64.97 | 76.27 |
| InfLoRA | $69.66^{\pm0.88}$ | $79.70^{\pm0.66}$ | $63.86^{\pm1.16}$ | $76.31^{\pm0.94}$ | $55.09^{\pm0.16}$ | $70.71^{\pm1.24}$ | 62.87 | 75.57 |
| **Ours** | $\mathbf{72.89^{\pm0.35}}$ | $\mathbf{81.34^{\pm0.64}}$ | $\mathbf{72.47^{\pm1.00}}$ | $\mathbf{81.55^{\pm1.28}}$ | $\mathbf{70.97^{\pm0.82}}$ | $\mathbf{80.61^{\pm0.67}}$ | **72.11** | **81.17** |

Table 5: Comparison of the performance of different CIL methods on TinyImageNet. We test the results of different methods under *three class orders* and report the mean and standard deviation.

| | T200-5T | | T200-10T | | T200-20T | | Average | |
| --- | --- | --- | --- | --- | --- | --- | --- | --- |
| | $A_{last}$ | Avg | $A_{last}$ | Avg | $A_{last}$ | Avg | $A_{last}$ | Avg |
| **Joint** | $71.64^{\pm0.15}$ | $77.43^{\pm0.56}$ | $71.54^{\pm0.19}$ | $78.28^{\pm0.64}$ | $71.99^{\pm0.08}$ | $79.23^{\pm0.47}$ | 71.72 | 78.31 |
| MEMO | $43.28^{\pm0.13}$ | $61.72^{\pm0.55}$ | $35.74^{\pm1.24}$ | $58.35^{\pm0.23}$ | $31.16^{\pm1.30}$ | $54.94^{\pm1.20}$ | 36.73 | 58.34 |
| FOSTER | $57.58^{\pm0.18}$ | $70.09^{\pm0.11}$ | $56.09^{\pm0.73}$ | $68.80^{\pm0.05}$ | $53.31^{\pm0.42}$ | $66.73^{\pm0.11}$ | 55.66 | 68.54 |
| MORE | $62.79^{\pm0.13}$ | $71.94^{\pm0.46}$ | $60.44^{\pm0.27}$ | $70.88^{\pm0.24}$ | $57.69^{\pm0.69}$ | $68.98^{\pm0.22}$ | 60.31 | 70.60 |
| ROW | $62.79^{\pm0.41}$ | $71.72^{\pm0.50}$ | $61.76^{\pm0.45}$ | $72.29^{\pm0.58}$ | $60.07^{\pm0.31}$ | $71.83^{\pm0.39}$ | 61.54 | 71.95 |
| TPL | $\mathbf{66.86^{\pm0.32}}$ | $\mathbf{75.04^{\pm0.63}}$ | $\mathbf{64.89^{\pm0.22}}$ | $\mathbf{74.67^{\pm0.53}}$ | $\mathbf{64.53^{\pm0.16}}$ | $\mathbf{74.08^{\pm0.47}}$ | **65.43** | **74.60** |
| LAE-Adapter | $63.12^{\pm0.34}$ | $72.29^{\pm0.70}$ | $60.04^{\pm0.87}$ | $70.71^{\pm1.36}$ | $55.44^{\pm0.95}$ | $67.54^{\pm1.62}$ | 59.53 | 70.18 |
| LAE-Prefix | $63.32^{\pm0.59}$ | $72.48^{\pm0.93}$ | $58.99^{\pm1.36}$ | $69.93^{\pm1.63}$ | $55.57^{\pm0.87}$ | $67.46^{\pm1.32}$ | 59.29 | 69.96 |
| LAE-LoRA | $63.58^{\pm0.23}$ | $72.63^{\pm0.76}$ | $59.57^{\pm1.29}$ | $70.69^{\pm1.26}$ | $55.45^{\pm1.08}$ | $67.36^{\pm1.81}$ | 59.53 | 70.23 |
| L2P | $60.91^{\pm0.53}$ | $70.03^{\pm0.68}$ | $56.39^{\pm0.61}$ | $68.47^{\pm0.23}$ | $52.51^{\pm0.81}$ | $65.59^{\pm0.75}$ | 56.60 | 68.03 |
| DualPrompt | $60.44^{\pm0.22}$ | $69.53^{\pm0.65}$ | $57.53^{\pm0.90}$ | $68.65^{\pm0.71}$ | $52.41^{\pm0.28}$ | $65.22^{\pm0.92}$ | 56.79 | 67.80 |
| CODA-Prompt | $61.98^{\pm0.31}$ | $71.42^{\pm0.32}$ | $58.44^{\pm0.45}$ | $69.91^{\pm0.78}$ | $54.80^{\pm0.56}$ | $67.57^{\pm0.25}$ | 58.40 | 69.63 |
| PASS | $63.33^{\pm0.37}$ | $72.44^{\pm0.44}$ | $60.62^{\pm0.15}$ | $71.20^{\pm0.47}$ | $57.40^{\pm0.64}$ | $69.20^{\pm0.76}$ | 60.45 | 70.94 |
| O-LoRA | $61.45^{\pm0.48}$ | $72.22^{\pm0.40}$ | $60.66^{\pm0.66}$ | $71.59^{\pm0.79}$ | $55.77^{\pm0.25}$ | $68.44^{\pm0.41}$ | 59.29 | 70.75 |
| EASE | $56.93^{\pm0.31}$ | $66.36^{\pm0.26}$ | $56.70^{\pm0.34}$ | $67.70^{\pm0.72}$ | $54.81^{\pm0.73}$ | $67.09^{\pm0.29}$ | 56.15 | 67.05 |
| InfLoRA | $60.32^{\pm0.29}$ | $70.14^{\pm0.52}$ | $56.43^{\pm0.35}$ | $68.36^{\pm0.15}$ | $51.49^{\pm0.38}$ | $64.80^{\pm0.32}$ | 56.08 | 67.77 |
| **Ours** | $\mathbf{64.42^{\pm0.67}}$ | $\mathbf{73.68^{\pm0.50}}$ | $\mathbf{64.36^{\pm1.11}}$ | $\mathbf{74.56^{\pm0.87}}$ | $\mathbf{63.62^{\pm0.24}}$ | $\mathbf{73.73^{\pm0.38}}$ | **64.13** | **73.99** |

Table 6: Comparison of the performance of different CIL methods on ImageNet380. We test the results of different methods under *three class orders* and report the mean and standard deviation.

| | I380-5T | | I380-10T | | I380-20T | | Average | |
|---|---|---|---|---|---|---|---|---|
| | $A_{last}$ | Avg | $A_{last}$ | Avg | $A_{last}$ | Avg | $A_{last}$ | Avg |
| **Joint** | $79.50^{\pm0.08}$ | $84.01^{\pm0.47}$ | $79.87^{\pm0.17}$ | $84.72^{\pm0.53}$ | $79.46^{\pm0.23}$ | $85.46^{\pm0.38}$ | 79.61 | 84.73 |
| MEMO | $51.22^{\pm0.89}$ | $66.40^{\pm1.33}$ | $49.79^{\pm0.76}$ | $67.98^{\pm1.76}$ | $51.83^{\pm0.81}$ | $68.51^{\pm1.64}$ | 50.95 | 67.63 |
| FOSTER | $62.93^{\pm0.59}$ | $74.65^{\pm1.16}$ | $64.29^{\pm0.42}$ | $74.15^{\pm1.86}$ | $63.10^{\pm0.61}$ | $73.31^{\pm1.58}$ | 63.44 | 74.04 |
| MORE | $72.75^{\pm0.18}$ | $80.63^{\pm0.44}$ | $69.26^{\pm0.55}$ | $78.44^{\pm0.76}$ | $66.80^{\pm1.03}$ | $76.41^{\pm1.34}$ | 69.60 | 78.49 |
| ROW | $73.12^{\pm0.24}$ | $80.63^{\pm0.56}$ | $72.09^{\pm0.22}$ | $80.61^{\pm0.36}$ | $70.95^{\pm0.32}$ | $80.45^{\pm0.47}$ | 72.05 | 80.56 |
| TPL | $\mathbf{76.88}^{\pm\mathbf{0.21}}$ | $\mathbf{82.69}^{\pm\mathbf{0.46}}$ | $\mathbf{75.32}^{\pm\mathbf{0.12}}$ | $\mathbf{81.79}^{\pm\mathbf{0.39}}$ | $\mathbf{74.95}^{\pm\mathbf{0.21}}$ | $\mathbf{81.45}^{\pm\mathbf{0.54}}$ | 75.72 | 81.98 |
| LAE-Adapter | $66.13^{\pm1.25}$ | $76.02^{\pm0.87}$ | $59.85^{\pm2.57}$ | $71.46^{\pm0.51}$ | $55.85^{\pm2.97}$ | $68.00^{\pm1.58}$ | 60.61 | 71.83 |
| LAE-Prefix | $70.02^{\pm0.73}$ | $78.67^{\pm0.43}$ | $64.55^{\pm0.96}$ | $75.14^{\pm0.25}$ | $59.82^{\pm0.59}$ | $71.49^{\pm0.42}$ | 64.80 | 75.10 |
| LAE-LoRA | $69.73^{\pm0.28}$ | $78.15^{\pm0.62}$ | $64.49^{\pm1.04}$ | $75.75^{\pm0.64}$ | $60.29^{\pm0.92}$ | $71.93^{\pm0.45}$ | 64.84 | 75.28 |
| L2P | $66.22^{\pm0.90}$ | $74.70^{\pm0.51}$ | $62.41^{\pm1.27}$ | $71.14^{\pm0.67}$ | $58.66^{\pm2.30}$ | $68.19^{\pm0.96}$ | 62.43 | 71.34 |
| DualPrompt | $65.54^{\pm0.91}$ | $75.58^{\pm0.97}$ | $62.86^{\pm1.11}$ | $74.40^{\pm0.53}$ | $61.70^{\pm1.39}$ | $73.28^{\pm0.58}$ | 63.37 | 74.42 |
| CODA-Prompt | $68.93^{\pm0.71}$ | $77.65^{\pm0.74}$ | $65.04^{\pm0.54}$ | $75.82^{\pm0.18}$ | $61.31^{\pm1.65}$ | $73.42^{\pm0.25}$ | 65.09 | 75.63 |
| PASS | $67.49^{\pm0.09}$ | $76.09^{\pm0.02}$ | $64.40^{\pm1.29}$ | $74.88^{\pm0.73}$ | $62.23^{\pm1.53}$ | $73.52^{\pm1.57}$ | 64.71 | 74.83 |
| O-LoRA | $59.95^{\pm0.44}$ | $75.98^{\pm0.02}$ | $58.28^{\pm1.13}$ | $72.09^{\pm0.86}$ | $50.78^{\pm1.13}$ | $69.76^{\pm0.87}$ | 56.34 | 72.61 |
| EASE | $64.88^{\pm0.49}$ | $72.71^{\pm0.77}$ | $64.40^{\pm0.27}$ | $73.78^{\pm0.55}$ | $62.70^{\pm0.44}$ | $73.63^{\pm0.56}$ | 63.99 | 73.37 |
| InfLoRA | $72.50^{\pm0.08}$ | $80.30^{\pm0.52}$ | $67.53^{\pm0.95}$ | $77.57^{\pm0.33}$ | $61.80^{\pm0.56}$ | $73.98^{\pm0.60}$ | 67.28 | 77.28 |
| **Ours** | $74.90^{\pm0.04}$ | $81.83^{\pm0.36}$ | $72.69^{\pm0.58}$ | $81.29^{\pm0.34}$ | $72.45^{\pm0.21}$ | $80.59^{\pm0.22}$ | 73.35 | 81.24 |

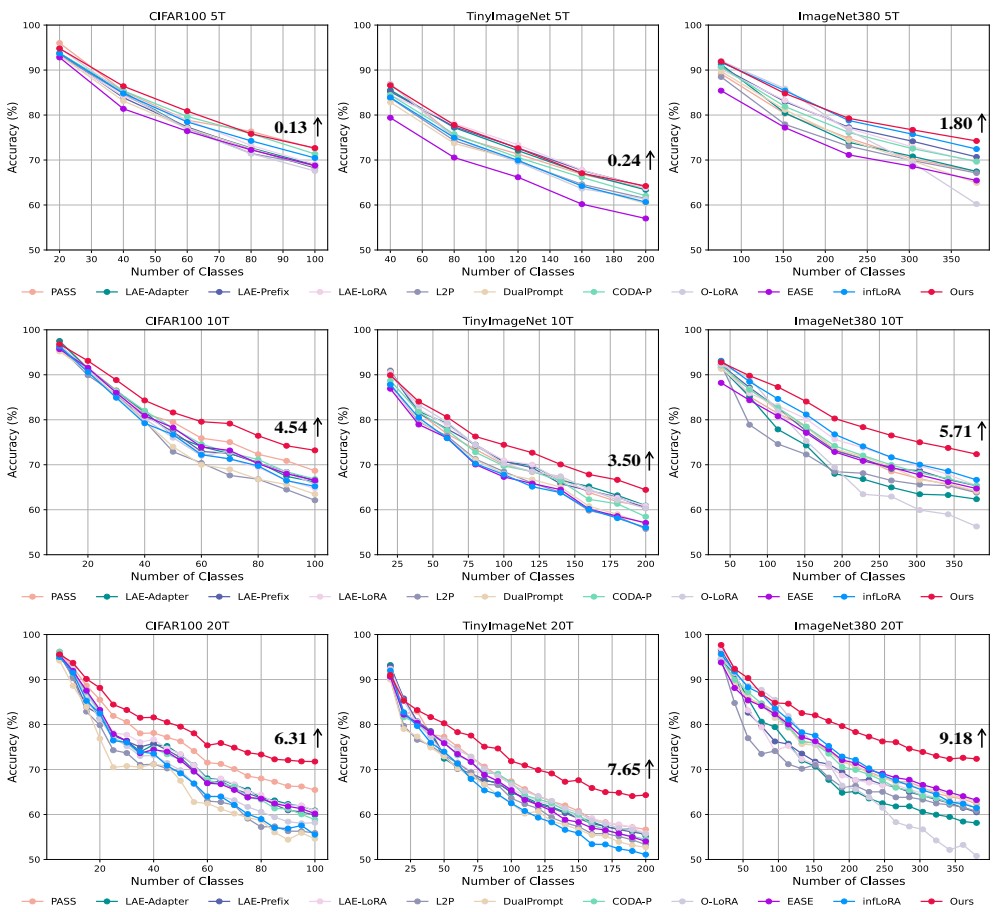

Figure 9: Results of accuracy curve on CIFAR100, TinyImageNet and ImageNet380.

## A.3 SIMILARITY BETWEEN LORAS FOR DIFFERENT TASKS

In Fig. 10. We plot the average cosine similarity between the different tasks of LoRA at the CIFAR100-10T setting. It can be seen that the natural orthogonality between the LoRA parameters of the different tasks is still exhibited without imposing additional constraints. The advantage of maintaining orthogonality between parameter spaces is that it lends itself to the ability to access different tasks directly through parameter fusion (Ilharco et al., 2022).

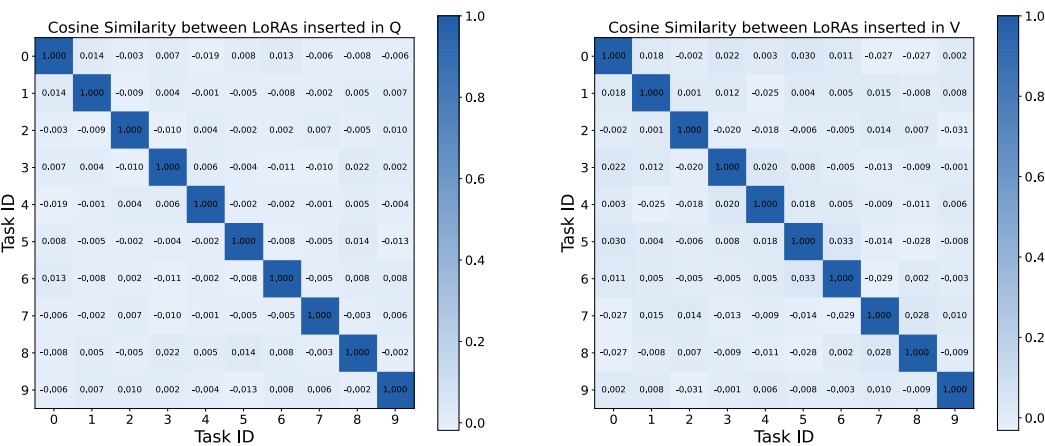

Figure 10: Visualization of cosine similarity between LoRAs for different task. The parameter spaces of the different task LoRAs maintain natural orthogonality with each other.

## A.4   FEATURE SPACE VISUALIZATION

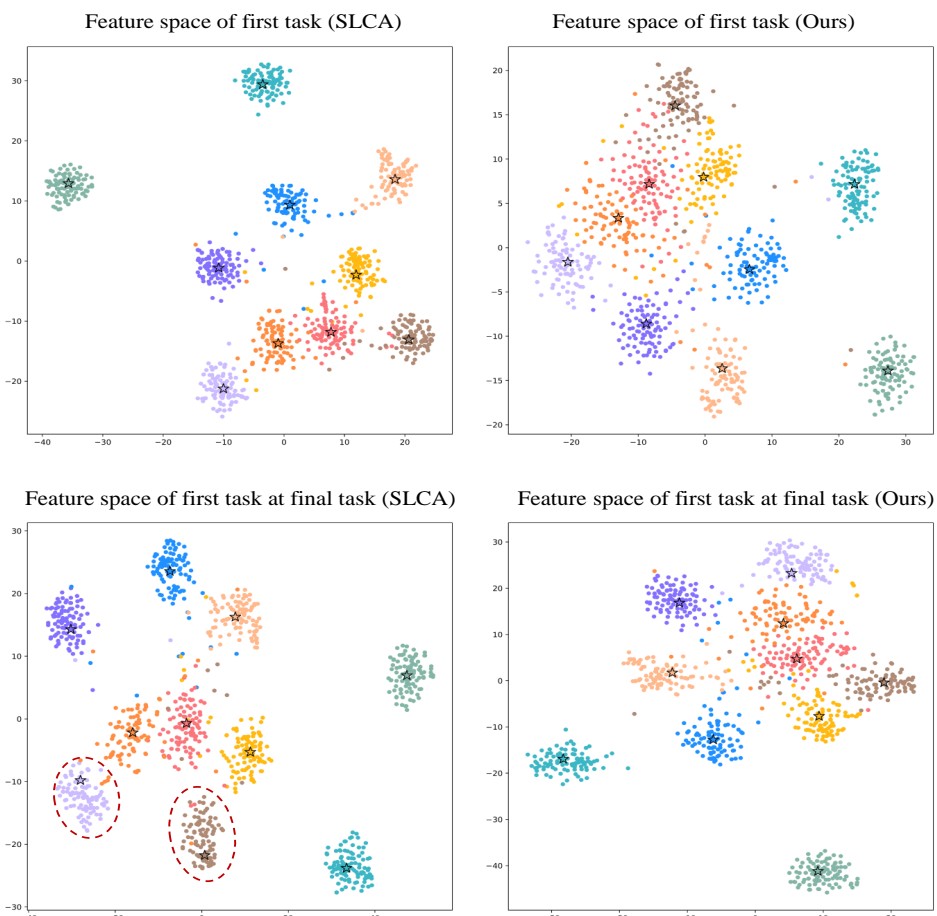

Figure 11: t-SNE visualization of the first task test dataset features after learned in the first stage and in the final stage. The mean feature of each class counted in training stage is denoted as ★. We highlight the phenomenon of feature drift with a red circle box.

In the training process, we calibrate the classifier by sampling pseudo-features and utilizing the per-class feature means and covariance matrices. We want the feature means of each class are as close as possible to the centers of their respective feature clusters and remain stable during subsequent training. However, as shown in Fig. 11, SLCA exhibits significant feature drift (we highlight in red circle box). This can lead to pseudo-features generated during the calibration stage to deviate from the true distribution and be confused with other classes, thus affecting the final performance. In contrast, our method mitigates feature drift by dynamically integrating the parameter spaces of both old and new tasks.

