# OpenReview forum: "DESIRE: Dynamic Knowledge Consolidation for Rehearsal-Free Continual Learning"
_ICLR.cc/2025/Conference — ICLR 2025 Conference Withdrawn Submission_

### Official Review · Reviewer_sTny · 2024-10-19

**Soundness:** 2
**Presentation:** 2
**Contribution:** 1
**Rating:** 3
**Confidence:** 4

**Summary:**

This paper focuses on the class-incremental learning problem using pretrained models. The authors designed a new LoRA-based rehearsal-free
method named DESIRE, integrating knowledge from old and new tasks with two post-processing modules. One to merge the LoRAs, the other to refine the classifier. The experiments show the superior performance over other methods.

**Strengths:**

1.	The paper is clearly written.

**Weaknesses:**

1.	The proposed LoRA merging paradigm seems to violate the protocol of supervised learning, which uses unlabeled test samples to perform the merging. It exposes the distribution of the test set to the model, making the learned merging coefficients better fits the test set.
2.	Sampling “pseudo-features” from the statistical information and use it to alleviate classifier bias is not novel (Such as _Semantic Augmentation_ in [1]).

[1] Class-Incremental Learning via Dual Augmentation, NIPS 2021

**Questions:**

See weaknesses.

---

### Official Review · Reviewer_4CZq · 2024-11-01

**Soundness:** 2
**Presentation:** 3
**Contribution:** 2
**Rating:** 5
**Confidence:** 5

**Summary:**

This paper proposes an incremental learning method based on LoRA (Low-Rank Adaptation), utilizing a greedy algorithm to retain only two LoRA sets, significantly reducing the burden of LoRA in handling a large number of tasks. Additionally, by retraining the classifier, the method avoids weight bias issues.

**Strengths:**

1.The paper is logically structured and rigorously written, making it easy for readers to follow.

2.The authors propose using a greedy algorithm to select two LoRA sets, thereby avoiding the significant storage overhead associated with retaining a large number of LoRA sets for numerous tasks

3.Retraining the classifier avoids the issue of weight bias in the classifier

**Weaknesses:**

1.Line 196 mentions that a simple initialization can allow different LoRAs to achieve orthogonality after training. Is it really possible to achieve this effect with just a simple initialization? Furthermore, this observation is only visualized on the CIFAR dataset, which can sometimes be inaccurate and subject to chance. Please provide a quantitative measure of orthogonality across different tasks and datasets.

2.At line 251, the greedy algorithm is used as a means to iteratively find the optimal solution by leveraging local optima. However, there are likely complex dependencies between LoRAs for different tasks. How can we ensure that local optima can substitute for the global optimum? Please explain why a greedy algorithm can be used to select LoRAs and provide some comparative experiments to demonstrate its effectiveness. Specifically, discuss the computational complexity of the greedy algorithm compared to other potential selection methods such as random selection, using all LoRAs, and selecting LoRAs based on their importance, etc. Additionally, how does the number of sets affect the performance?

3.Are the results in Table 1 reproduced by the authors? There seem to be several inconsistencies compared to the results reported by the original authors. Please provide a brief discussion of their reproduction process, including any differences in implementation or experimental setup that might account for discrepancies.

4.The paper lacks the listing of hyperparameters, which makes it difficult for subsequent readers to reproduce the results. Please provide a detailed ablation study on key hyperparameters, showing how they affect the model's performance.

5.Experiments on ImageNet1k are not listed. Some incremental learning methods perform poorly on ImageNet1k. If time permits, please provide these experiments (it will not reduce the score).

6.The method of retraining the classifier has been used in previous papers. Please provide a table or figure directly comparing the classifier retraining approach with DER and any other relevant methods, showing both similarities and differences in methodology and performance across different datasets or task settings.. (below)

[1] DER: Dynamically Expandable Representation for Class Incremental Learning

**Questions:**

1.Line 196 mentions that a simple initialization can allow different LoRAs to achieve orthogonality after training. Is it really possible to achieve this effect with just a simple initialization?

2 How can we ensure that local optima can substitute for the global optimum?

3how does the number of sets affect the performance?( lora sets)

4.Are the results in Table 1 reproduced by the authors? There seem to be several inconsistencies compared to the results reported by the original authors. (above)

---

### Official Review · Reviewer_gdEg · 2024-11-01

**Soundness:** 3
**Presentation:** 3
**Contribution:** 3
**Rating:** 6
**Confidence:** 4

**Summary:**

This paper proposes a novel rehearsal-free continual learning DESIRE approach for dynamic knowledge consolidation.  It involves the merging of parameters from previous and current tasks using LoRA-based updates, which allows the model to maintain stability and adapt to new tasks without the need for rehearsing old tasks. To tackle the bias towards new classes, the method refines decision boundaries through pseudo-feature generation, helping the model better generalize across all learned tasks.

**Strengths:**

1. Rehearsal-Free Continual Learning: Unlike many existing methods that rely on rehearsing previous tasks, DESIRE operates without the need for storing past data, thus reducing memory overhead.
2. Efficient Utilization of Unlabeled Data: The method efficiently uses a small subset of unlabeled data to learn merging coefficients, which is particularly innovative and resource-efficient.
3. DESIRE demonstrates superior performance on multiple benchmark datasets compared to other state-of-the-art rehearsal-free methods.

**Weaknesses:**

1. In your paper, you mentioned comparing your method with "iCARL and DER." However, I couldn't find the corresponding results in the paper.
2. Regarding generalizability, I have a concern: in your method, the backbone is pre-trained on ImageNet 1K and kept fixed during training. I believe the distribution of CIFAR-100 and Tiny ImageNet is quite similar to ImageNet 1K. Therefore, during continual learning, there might not be many new features to learn. However, if you use a more diverse dataset that contains features different from ImageNet, can you still keep the pre-trained backbone fixed? Or is it possible to train the backbone continually with your method?

**Questions:**

See weaknesses.

---

### Official Review · Reviewer_B9hG · 2024-11-09

**Soundness:** 2
**Presentation:** 3
**Contribution:** 2
**Rating:** 3
**Confidence:** 5

**Summary:**

The topic of this paper is about rehearsal-free continual learning (CL) with LoRA parameter-efficient fine-tuning techniques and pre-trained models. The authors propose an efficient DESIRE framework, which consists of a LoRA parameters merging strategy for representation consolidation and a decision boundary refinement scheme for classifier bias alleviating. The proposed method has been evaluated on several benchmarks.

**Strengths:**

+ The paper is well-written and easy to follow.
+ The proposed method achieves performance gain on several benchmarks.
+ Exploring the effect of the gap between the pre-trained data and data of incremental learning tasks to CL is interesting.

**Weaknesses:**

- The work HiDe-PET[a] has employed different PEFT techniques (e.g., LoRA, adapter) in rehearsal-free CL scenarios. The experimental comparison to HiDe-PET are not include in this paper. As described in the paper[a], HiDe-LoRA achieves SOTA performance on several  CL benchmarks. For example, the performance of HiDe-LoRA on split-CIFAR100 is FAA=76.46% and CAA=84.89%. The performance of  DESIRE on CIFAR100-10T is FAA=72.47% and CAA=81.55%. To better show the SOTA performance of the proposed method, a performance and memory cost analysis about these methods on classical benchmarks is necessary.

[a] HiDe-PET: Continual Learning via Hierarchical Decomposition of Parameter-Efficient Tuning, arxiv2024

- Conduct experiments on the CL benchmarks with a large gap between the pre-trained data and incremental learning data. The work [b] has also pointed out the data overlapping problem between the pre-trained data and CL data, and they proposed four benchmarks (ImageNet-A, ObjectNet, OmniBenchmark and VTAB) to alleviate this problem. The proposed method can be evaluated on such benchmarks to further show the advantages.

[b] Revisiting Class-Incremental Learning with Pre-Trained Models: Generalizability and Adaptivity are All You Need, arxiv2023

- Concerns about the idea of only maintaining two sets of LoRA parameters. When the data of a new task arrives, the model should adaptively decide whether to learn a new set of PEFT parameters or not (e.g. the work [c] described the prompts parameter). If the difference between the new task and previously learned tasks is large, the newly learned LoRA parameters should be maintained. If the new task is similar as some of previously learned tasks, the newly learned LoRA parameters can be merged with that of such old tasks. If the new task is almost same as previously learned tasks, the new LoRA parameters are not needed. Can the authors make a discussion about this key point? Can maintaining only two sets of LoRA parameters deal with the cross-domain CL case well? In a cross-domain CL case (e.g., VTAB benchmark in [b]), each new task is very different from the previously learned tasks.

[c] LW2G: Learning Whether to Grow for Prompt-based Continual Learning, arxiv2024

**Questions:**

My major concerns come from the experimental results about related works and complex CL benchmarks, the concerns about the LoRA merging method. The details are included in the above weaknesses.

---

### Note · Authors · 2024-11-28

I have read and agree with the venue's withdrawal policy on behalf of myself and my co-authors.